



# Spatial filtering in a 6D hybrid-Vlasov scheme for alleviating AMR artifacts: a case study with Vlasiator, versions 5.0, 5.1, 5.2.1

Konstantinos Papadakis[1], Yann Pfau-Kempf[1], Urs Ganse[1], Markus Battarbee[1], Markku Alho[1], Maxime Grandin[1], Maxime Dubart[1], Lucile Turc[1], Hongyang Zhou[1], Konstantinos Horaites[1], Ivan Zaitsev[1], Giulia Cozzani[1], Maarja Bussov[1], Evgeny Gordeev[1], Fasil Tesema[1], Harriet George[1], Jonas Suni[1], Vertti Tarvus[1], and Minna Palmroth[1,2]

[1]University of Helsinki, Department of Physics, Helsinki, Finland
[2]Finnish Meteorological Institute, Space and Earth Observation Centre, Helsinki, Finland

**Correspondence:** Konstantinos Papadakis (konstantinos.papadakis@helsinki.fi)

**Abstract.** Numerical simulation models that are used to investigate the near-Earth space plasma environment require sophisticated methods and algorithms together with high computational power. Vlasiator 5.0 is a hybrid-Vlasov plasma simulation code that is able to perform 6D (3D in ordinary space and 3D in velocity space) simulations using Adaptive Mesh Refinement (AMR). In this work we describe a side effect of using AMR in Vlasiator 5.0 where the heterologous grid approach creates resolution induced discontinuities due to the different grid resolution levels. The discontinuities cause spurious oscillations in the electromagnetic fields that alter the global results. We present and test a spatial filtering operator for alleviating this artifact without significantly increasing the computational overhead. We demonstrate the operator's use case in large 6D AMR simulations and evaluate its performance with different implementations.

## 1 Introduction

Investigation of the near-Earth space plasma environment benefits from numerical simulation efforts, which can model plasma effects on global scales compared to physical observations that are inherently local in space and time (Hesse et al., 2014). Vlasiator (Palmroth et al., 2018) is a hybrid-Vlasov plasma simulation code that models collisionless plasma by solving the Vlasov-Maxwell system of equations for ion particle distribution functions on a six dimensional Cartesian mesh, with 3 dimensions representing position and 3 velocity space. The Vlasov Equation (1) is a form of the Boltzmann equation that neglects the collisional term to only account for electromagnetic interactions:

$$\frac{\partial f}{\partial t} + \boldsymbol{v} \cdot \frac{\partial f}{\partial \boldsymbol{r}} + \frac{q}{m}\left(\boldsymbol{E} + \boldsymbol{v} \times \boldsymbol{B}\right) \cdot \frac{\partial f}{\partial \boldsymbol{v}} = 0. \tag{1}$$

Here, $f(\boldsymbol{r}, \boldsymbol{v}, t)$ represents the phase space density of a species of mass $m$ and charge $q$, where $\boldsymbol{r}$ is position, $\boldsymbol{v}$ is velocity and $t$ is time. $\boldsymbol{E}$ and $\boldsymbol{B}$ stand for the electric and magnetic fields respectively. Vlasiator couples the Vlasov equation for ions with the electromagnetic fields through Maxwell's equations under the Darwin approximation which in practice eliminates the displacement current in the Ampère equation to get rid of electromagnetic wave modes and enable longer timesteps. This leads to the Ampère, Faraday and Gauss laws taking the form:



$$j = \frac{\nabla \times \boldsymbol{B}}{\mu_0} \qquad (2)$$

$$\nabla \times \boldsymbol{E} = -\frac{\partial}{\partial t}\boldsymbol{B} \qquad (3)$$

$$\nabla \cdot \boldsymbol{B} = 0. \qquad (4)$$

The system is closed using Ohm's law in the form

$$\boldsymbol{E} = -\boldsymbol{V_i} \times \boldsymbol{B} + \frac{1}{\rho_q}\boldsymbol{j} \times \boldsymbol{B} - \frac{1}{\rho_q}\nabla \boldsymbol{P_e}, \qquad (5)$$

where $\boldsymbol{j}$ is the current density, $\mu_0$ is the permeability of free space, $\boldsymbol{V_i}$ is the ion bulk velocity, $\rho_q$ is the charge density, and $\boldsymbol{P_e}$ is the electron pressure tensor.

In its implementation, Vlasiator stores a 3D velocity grid in each spatial grid cell, which requires significant memory for
large simulations. This leads to simulation results that are free from sampling noise, unlike simulations that employ stochastic particle representation methods such as Particle-In-Cell (PIC) codes (Lapenta, 2012). While ion kinetics are resolved, Vlasiator models the electron population as a charge-neutralizing background fluid, as typical in hybrid-kinetic approaches, to keep computational cost down. Vlasiator employs a sparse velocity space representation (von Alfthan et al., 2014), where the parts of the velocity distribution function below a specific threshold are neither stored nor propagated. The electromagnetic fields
are coupled to the Vlasov solver by taking velocity moments of the distribution function (density, flow velocity, pressure) and feeding them into Maxwell's Equations (2)-(5) which are then solved through a constrained transport upwind method described in Londrillo and del Zanna (2004).

Vlasiator's core is made up of two separate solvers, the field solver and the Vlasov solver. The Vlasov solver solves the Vlasov equation in two steps using Strang splitting (Palmroth et al., 2018), namely spatial translation and acceleration in
velocity space, using a semi-Lagrangian scheme based on the SLICE-3D method described in Zerroukat and Allen (2012). Vlasiator has been employed in a range of studies regarding Earth's foreshock formation (Turc et al., 2019; Kempf et al., 2015), ionospheric precipitation (Grandin et al., 2019) and magnetotail reconnection (Palmroth et al., 2017), for example.

Most scientific studies of Vlasiator have been limited to 5 dimensions (two spatial and three velocity dimensions) due to the large computational requirements. In Vlasiator 5.0, Adaptive Mesh Refinement (AMR) has been applied to enable the
simulation of 6D configurations. With the use of AMR the Vlasov solver uses the highest spatial resolution available only in regions of high scientific interest. Regions of less interest are solved at a lower spatial resolution. Since Vlasiator needs to store a velocity distribution function for every simulation cell, which is numerically described by a 3D grid, the memory requirements for 6D simulations are extreme. The AMR functionality previously added in Vlasiator 5.0 manages to alleviate the computational burden by reducing the effective cell count in a 6D simulation. Thus the use of AMR is necessary for
Vlasiator to venture into exploring the near-Earth space plasma in 6D.

In this work, we demonstrate the use of low pass filtering in Vlasiator to help eliminate artifacts caused in numerical simulations using AMR. The structure of this publication is such that in Chapter 2 we provide an insight to Vlasiator's grid topology as well as the heterologous grid coupling mechanism to exchange the required variables between its two core solvers.





Moreover, we describe the staircase effect, an artifact caused by the grid coupling process in AMR runs. In Chapter 3 we give
a brief summary of low pass filtering and propose our method for alleviating the staircase effect in Vlasiator. In Chapter 4, we
demonstrate simulation results and evaluate the performance of two different implementations of the filtering operator. Further
discussion and conclusions are presented in the final chapters of this work.

## 2  Heterologous Grid Structure in Vlasiator

Vlasiator's field solver uses the FsGrid library (Palmroth and the Vlasiator team, 2020) to store field quantities, plasma moments
and propagate the electromagnetic fields. FsGrid is a Message Passing Interface (MPI) aware library that uses ghost cell
communication to make data available to neighboring processes in the simulation domain. The electromagnetic field update is
not an expensive operation in Vlasiator and thus requires no AMR, even in a spatially 3D high-resolution setup. Hence, the
spatial resolution for FsGrid is kept uniform in the simulation domain. FsGrid uses a block decomposition scheme to evenly
distribute the simulation cells over the MPI tasks and for the rest of the simulation this is kept constant when the number of MPI
tasks remains the same. On the other hand, velocity distribution function translation and acceleration are the most expensive
operations in Vlasiator. Quantities of the Vlasov solver are discretized using DCCRG (Honkonen et al., 2013), (Honkonen,
2022), a parallel grid library that supports cell-based AMR. Hence Vlasiator employs a heterologous AMR scheme, where the
field solver and Vlasov solver operate on separate grids and with different spatial resolution depending on the local DCCRG
AMR level, with the field solver operating at the highest resolution allowed by DCCRG.

Currently for Vlasiator's 6D simulations, the refinement levels are manually chosen. Regions of interest closer to the Earth
like the bow shock and the magnetotail reconnection site operate at the maximum spatial resolution, whereas lower spatial
resolution is used for regions of less interest such as the inflow and outflow boundaries. In Figure 1 a typical configuration
of the AMR levels in a 6D Vlasiator simulation is depicted. Dynamically adjusting the AMR levels based on physical criteria
during run-time is under development and the subject of a future study.

### 2.1  Grid Coupling

DCCRG operates on a base refinement level and each successive refinement level has twice the resolution of the previous
one. At the highest refinement region there is a 1-to-1 match between the field solver and DCCRG's cells. However, the
electromagnetic fields and plasma propagation and acceleration are inherently dependent upon each other and thus a coupling
process takes place during every simulation timestep. The coupling scheme is illustrated in Figure 2. The Vlasov solver at the
end of every timestep feeds moments into the field solver grid. In regions where the 1-to-1 match is not fulfilled, one set of
moments is communicated to all FsGrid cells which occupy the same volume as the underlying DCCRG cell. The field solver
then propagates the fields and communicates those back to the Vlasov solver before the next time step begins. In mismatching
regions, the field solver grid is fed uniform input in all the cells that are children of a lower resolution DCCRG cell and
later the parent DCCRG cell is fed an averaged value of all the higher resolution corresponding FsGrid cells. The association
between the two grids is calculated during the initialization and after every load balancing operation where the Cartesian spatial



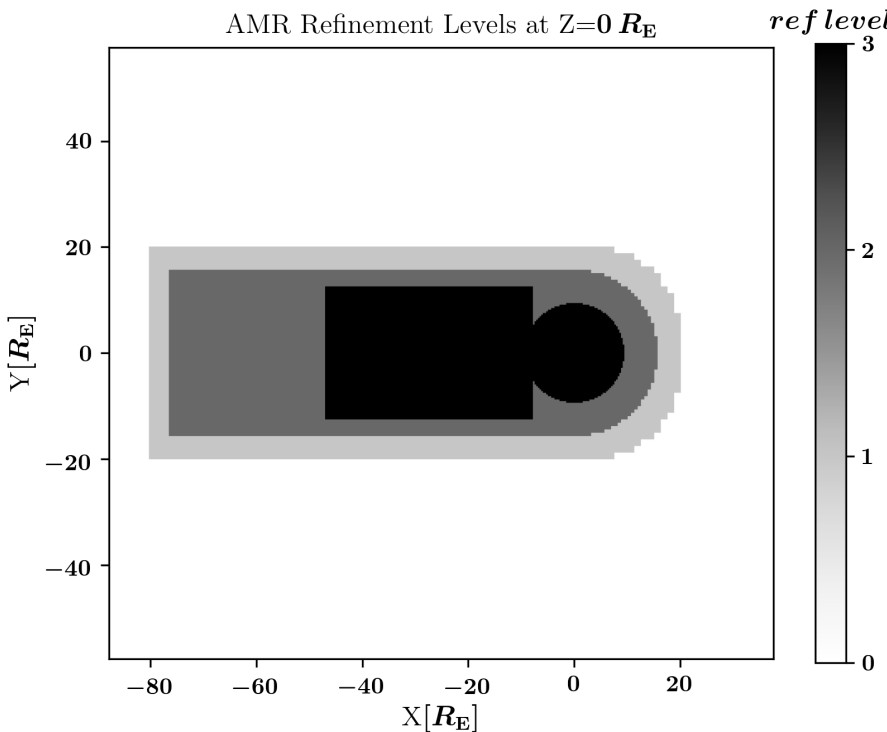

**Figure 1.** Equatorial distribution of the AMR refinement levels on the Vlasov grid in a 3D AMR simulation. Refinement level n corresponds to $2^n$ times the base resolution. Darker regions closer to the Earth (n=3) are solved using the highest spatial resolution used in the simulation.

decomposition scheme over different MPI tasks changes for the DCCRG grid. When no AMR is used the two solvers operate on the same spatial resolution and thus there is a 1-to-1 grid match making the coupling scheme trivial.

## 2.2 Staircase Effect

In 6D AMR simulations the 1-to-1 grid matching is restricted to only the highest refinement regions where both solvers operate at the highest spatial resolution. In less refined regions, the Vlasov solver cells span multiple field solver cells and the grids mismatch. If the trivial coupling scheme described in the previous subsection is maintained, the field solver is subject to discontinuous plasma moment input at the Vlasov grid cell interfaces, which can be seen in Figures 3(a) and (b), as an effect that we dub the staircase effect. The discontinuities caused by the staircase effect lead to the development of unphysical oscillations in the field quantities on the field solver grid. The oscillations can be observed in the profiles demonstrated in Figures 3(d) and (f). Those oscillations can act as a source of spurious wave excitation and propagate artifacts in the whole simulation domain as visible in Figures 3(c) and (e), where artifacts have propagated downstream from the bow shock of the global magnetospheric simulation, causing significant distortion of the physics in the nightside magnetosheath and lobes.





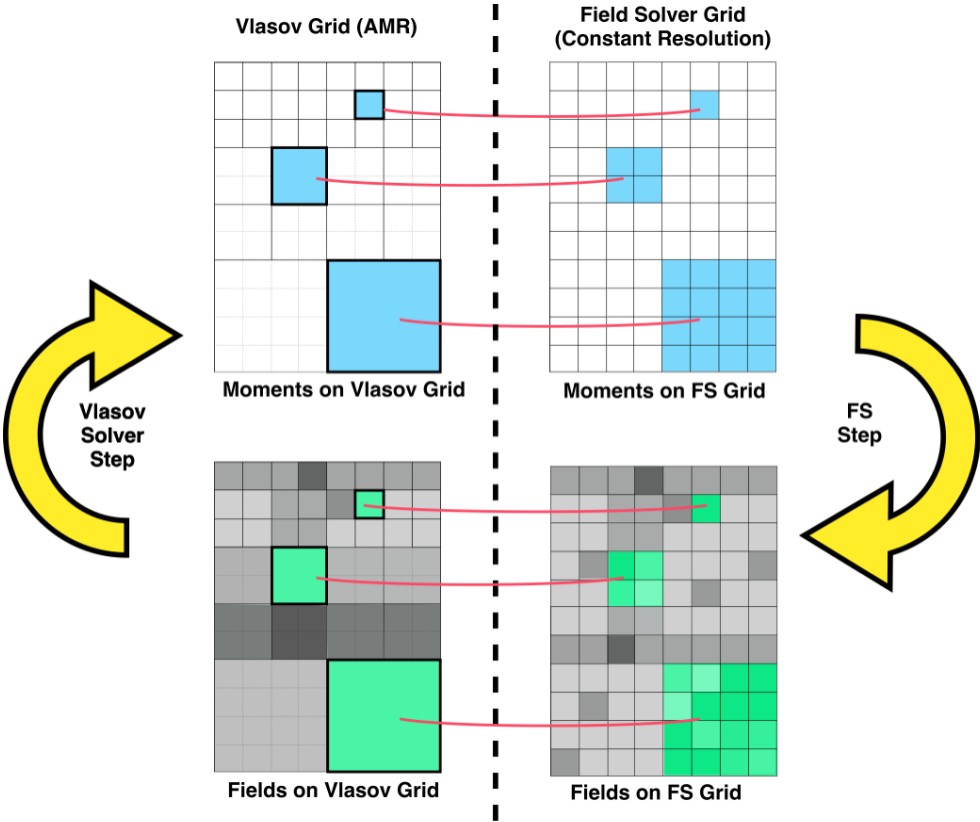

**Figure 2.** Schematic of the grid coupling procedure. Moments evaluated on the Vlasov grid are copied over to FsGrid. When there is a grid mismatch, one DCCRG cell copies over its values to many FsGrid cells. Then the fields are propagated and finally fed back to the Vlasov grid. In areas where the grids mismatch, multiple FsGrid cells are averaged to get the value for the corresponding DCCRG cell.

## 3 Method

### 3.1 Low Pass Filtering

Low pass filtering is a well known tool from digital signal processing theory that effectively attenuates unwanted high frequency parts of the spectrum. The boxcar filter is a Finite Impulse Response low pass filter, and its frequency response is given by

$$H\left(\nu\right) = \frac{\sin\left(\pi\nu N\right)}{\sin\left(\pi\nu\right)} e^{-j\pi\nu(N-1)}, \tag{6}$$

where $\nu$ here is the frequency, $H$ is the frequency response and $N$ is the length of the kernel. Boxcar filters are usually cascaded with other boxcars in an attempt to reduce the high side-lobes in their frequency response (Roscoe and Blair, 2016). Techniques

like low pass filtering are not limited to the time domain; they can be applied to other dimensions like ordinary space and find wide application in fields like image processing (Cook, 1986) and numerical modeling (Vay and Godfrey, 2014).



## 3.2 Spatial Filtering

In this work we present two implementations of the spatial filtering operator used in Vlasiator to smooth out the discontinuities in AMR simulations that are illustrated in Figures 3(a,b). First, in Vlasiator 5.1 the filtering operator is realized by a 3-dimensional 27-point (3 points per spatial dimension) boxcar kernel with equally assigned weights. The kernel operates in position space ($\boldsymbol{r}$), and is passed over the field solver grid cells immediately after the coupling process finishes transferring Vlasov moments to the field solver grid. The filtering operator is only applied when there is a grid mismatch between the two solvers, so it is not used in the highest refinement level where the two solvers operate at the same spatial resolution. The number of times the operator is applied is not constant but linearly depends on the refinement ratio between the two grids. Each filtering operator pass attenuates the high frequency signals on the field solver grid and smooths out the discontinuities shown in Figures 3(a,b). Larger refinement ratios between the two grids give rise to spatially larger discontinuities, thus requiring more filtering passes in order to smooth the discontinuity. In practice, treating the finest to coarsest levels with 0,2,4,8,... passes of the boxcar kernel has proved to alleviate the discontinuities illustrated in Figures 3(a,b).

At the end of each pass of the filtering scheme, a ghost cell communication update is required for the respective FsGrid structure prior to continuing on to the next pass. This manifests as a performance penalty, since the ghost communication is a global process involving all MPI tasks in the simulation. From the associative properties of the convolution operation, where $B$ is a boxcar kernel, $T$ is a triangle kernel, $g$ is a function and $\circledast$ is the convolution operator:

$$(g \circledast B) \circledast B = g \circledast (B \circledast B) = g \circledast T \tag{7}$$

two boxcar passes are the equivalent of a single triangular kernel pass, so in Vlasiator 5.2.1 we update our filtering operator to a 3-dimensional 125-point triangle kernel. We choose the 3D triangle kernel as the convolution of a boxcar kernel with itself to match the frequency characteristics of the boxcar operator in half the number of passes as shown in Figure 4. The triangle kernel is used in the same fashion as the boxcar kernel, however since this time half the number of passes per refinement level are required to achieve the same amount of smoothing the triangular kernel operator performs half the amount of ghost cell communications. A graphical representation of the filtering kernels is illustrated in Figure 5. In Algorithm 1 we demonstrate the two implementations of the spatial filtering algorithm with a 1D pseudo-code example. The modified coupling scheme is demonstrated in Figure 6, where now it is supplemented with the filtering mechanism in contrast to Figure 2.

## 4 Results

To demonstrate the effect of the spatial filtering employed in Vlasiator 5.1 to alleviate the AMR discontinuities, we create a simple configuration with an artificial step in the middle of a 3D simulation box. The step shown in Figure 7 poses a discontinuity in the otherwise smooth mass density. Similar steps are created during AMR simulations in regions where there is no 1-to-1 match between the Vlasov and field solver grid cells. We apply the boxcar filtering operator for an increasing number of times to evaluate its performance and the results are illustrated in Figure 7. When the filtering operator is cascaded with itself, its effect becomes more significant in damping high frequency signals as shown in Figure 4. In Figure 7, we only





---

**Algorithm 1** Filtering Operator Pseudo-code

---

1: **if** method=="boxcar" **then**
2:   $kernelWidth \leftarrow 1$
3:   $kernelWeights \leftarrow [1,1,1]$
4:   $sum \leftarrow sum(kernelWeights)$
5: **else if** method=="triangle" **then**
6:   $kernelWidth \leftarrow 2$
7:   $kernelWeights \leftarrow [1,2,3,2,1]$
8:   $sum \leftarrow sum(kernelWeights)$
9: **end if**
10: $swapGrid \leftarrow momentsGrid$
11: **for** $pass = 0,1,\ldots maxPasses$ **do**
12:   **for** $i = 0,1,\ldots momentsGrid.GridSize$ **do**
13:     $refLevel \leftarrow momentsGrid.getRefLevel(i)$
14:     **if** $pass >= momentsGrid.numPasses(refLevel)$ **then**
15:       continue
16:     **end if**
17:     $swapGrid[i] \leftarrow 0.0$
18:     **for** $j = -kernelWidth\ldots kernelWidth$ **do**
19:       $ii \leftarrow i+j$
20:       $swapGrid[i] += momentsGrid[ii] * kernelWeights[kernelWidth+j]/sum$
21:     **end for**
22:   **end for**
23:   $momentsGrid \leftarrow swapGrid$
24:   $momentsGrid.updateGhostCells()$
25: **end for**

---

show the effect of the boxcar operator since the results are identical to using the triangular kernel operator, as shown by the
frequency response of the two kernels depicted in Figure 4.

Furthermore, we demonstrate results of the boxcar filtering operator in a large magnetospheric production scale run with four
AMR levels as illustrated in Figure 8. In Figure 8(a) a colour map of the mass density on the field solver grid is depicted with
the filtering operator disabled. In Figure 8(b) the boxcar filtering operator is used and the AMR levels are treated, from finest
to coarsest, with 0,2,4, and 8 passes respectively. Figures 8(c) and 8(e) show the electric field and magnetic field magnitudes
simulated with filtered moments on FsGrid. The profiles demonstrated in Figures 8(d) and 8(f) are sampled respectively along
the dashed paths in panels (c) and (e) of Figure 8.

The simulation used to evaluate the filtering operator models a three dimensional space around Earth in the Geocentric Solar
Magnetic (GSM) coordinate system with no dipole tilt. The modeled space extends from $-560000\,\text{km}$ to $240000\,\text{km}$ in the



X dimension, from $-368000\,\mathrm{km}$ to $368000\,\mathrm{km}$ in both Y and Z dimensions and is represented by a $100 \times 92 \times 92$ Cartesian
mesh with $\Delta r = 8000\,\mathrm{km}$ at the lowest refinement level and with $\Delta r = 1000\,\mathrm{km}$ at the highest refinement level. Each spatial
cell contains a velocity space with a 3D grid with $\Delta v = 40\,\mathrm{km/s}$. The solar wind is modeled with proton density $n = 7\,\mathrm{cm}^{-3}$,
temperature $T = 0.5 \times 10^6\,\mathrm{K}$ and solar wind speed with $V_x = -1000\,\mathrm{km/s}$. The interplanetary magnetic field points mostly
southward with $\boldsymbol{B} = [-0.5, 0, -20]\,\mathrm{nT}$.

### 4.1 Performance Overhead

Care has to be taken to keep the performance overhead of the filtering operator small. The boxcar operator is applied at every
simulation timestep and makes use of a duplicate FsGrid structure of the Vlasov moments since the filtering cannot happen
in place. The boxcar operator uses OpenMP threading to parallelize the filtering over the local domain of each MPI task. In
the table below we report the extra memory needed for the filtering operator and the time spent filtering the Vlasov moments
during the grid coupling process for the production 6D AMR run shown in Figure 8.

| | |
|---|---|
| **Memory used for the simulation [GB]** | 11000 |
| **Memory used for the boxcar filtering [GB]** | 15.68 (0.14%) |
| **Time spent in simulation [s]** | 50410 |
| **Time spent in boxcar filtering [s]** | 2818 (6%) |

**Table 1.** Profiling statistics for the boxcar filtering operator.

From Table 1 we see that the boxcar filtering operator used in Vlasiator 5.1, amounts to $6\%$ of the total simulation time for
a production run like the one in Figure 8. To evaluate the performance improvement of the 5-stencil triangle kernel implementation, used in Vlasiator 5.2.1, we set up smaller tests and compare the two methods. The triangle kernel operates in the same
way but only needs half the numbers of passes to achieve proper smoothing so we treat the finest to coarsest levels with 0,1,2,4
passes. We report the results in Table 2. While both approaches require the same amount of memory, the time spent by the

| | |
|---|---|
| **Memory used for the simulation [GB]** | 250 |
| **Memory used for the boxcar operator [GB]** | 0.035 (0.014%) |
| **Memory used for the triangle operator [GB]** | 0.035 (0.014%) |
| **Time spent in simulation [s]** | 1260 |
| **Time spent by the boxcar operator [s]** | 147 (11.8%) |
| **Time spent by the triangle operator [s]** | 87 (7.0%) |

**Table 2.** Comparison of the two filtering kernels' performance.

triangular operator amounts to $59\%$ of that spent by the boxcar operator.





Both the boxcar and the triangular filtering operators are three-dimensional spatial convolutions that can be expressed as three one-dimensional convolutions (Birchfield, 2017). This is known as kernel separability and can improve the performance of the two operators significantly. Formally, the use of a separable kernel instead of a three-dimensional one, would reduce the complexity from $\mathcal{O}(Nx \times Ny \times Nz \times d^3)$, where $Nx, Ny, Nz$ are the dimensions of the simulation mesh and $d$ is the dimension size of the 3D kernel, to $\mathcal{O}(3 \times Nx \times Ny \times Nz \times d)$. We modify our implementations of the 3D boxcar and triangle operators to take advantage of the kernel separability property and test their performance using the same configuration used to produce the results in Table 2. We demonstrate the performance statistics of all four methods in Figure 9.

While the separable operators should theoretically lead to a significant performance gain, in practice the one-dimensional operators are slower than their three-dimensional counterparts. This is due to the fact that an interim ghost-update communication needs to take place after every pass done by the operators, and, since the one-dimensional implementations require more mesh traversals per pass, they end up spending more time in updating their ghost cell values. Additionally, we note that kernel separability does not hold if the stencil is altered, for example when part of the kernel covers the highest refinement level.

## 4.2 Moment Conservation

The filtering operator as described above is not conservative at the interface of adjacent refinements levels. However, this is not a cumulative effect since moments on FsGrid, used to propagate the electromagnetic fields, are provided to the field solver by the Vlasov solver at each timestep and they are not copied back from the FsGrid. Furthermore, the moment conservation is also violated due to numerical precision round-off errors during the filtering passes. The amount by which the moments are not conserved depends on the number of filtering passes and on the number of cells in a given simulation. We measure the relative difference in mass density caused by the filtering operator in the simulations presented in this work and find it to always remain below $10^{-5}$ which we deem acceptable given the non-cumulative nature of the filtering operation.

## 5 Discussion

The first 6D simulations with Vlasiator 5.0 would not be possible without the use of AMR. However the heterologous grid structure combined with the grid coupling mechanism in Vlasiator create artifacts in simulations using AMR that alter the global physics. In this work we report on a new development employing spatial filtering in the hybrid Vlasov code Vlasiator, versions 5.1 and 5.2.1, in order to alleviate the staircase effect created due to the heterologous AMR scheme used in 6D simulations. Based on the results of this study the use of a linearly increasing number of passes per refinement level minimizes the aliasing effect at the Vlasov–field-solver grid interfaces. Treating the finest to coarsest levels with 0, 2, 4, 8, . . . passes of the boxcar filter has proved to alleviate the staircase effect satisfactorily, as can be seen in Figures 8(a,b). As a result the electric and magnetic field magnitude profiles in Figures 8(d,f) show none of the oscillatory behavior caused by the staircase effect in contrast to those demonstrated in Figures 3(d,f). Since the filtering operator is applied at every simulation timestep, it has to be well optimized so that it does not increase the computational overhead significantly. From Table 1 we see that the filtering in a production simulation amounts to 6% of the computational time, which we deem significant. To improve the



filtering performance we develop a three dimensional 5-point stencil triangle kernel in Vlasiator 5.2.1, which is equivalent at alleviating the staircase effect but only needs half the number of passes per refinement level. We test the triangle kernel on a smaller simulation and report on its performance in Table 2. The triangle kernel provides a $41\%$ performance improvement over the boxcar approach and thus we estimate that in a similar simulation to the one shown in Figure 8 it would amount to $3.5\%$ of the total simulation time, which we deem acceptable. The improved performance is a combination of both halving the ghost cell updates needed for the triangular kernel operator as well as reducing the operations needed since the wider kernel operator requires half the number of passes compared to the boxcar operator. Furthermore, we evaluate the performance gain acquired by exploiting the filter separability property of the two filtering operators and conclude that the separable kernels in fact perform worse in the context of Vlasiator than their 3D counterparts as they are hindered by the higher number of ghost cell updates they require. Another approach to improve the performance of the filtering methods would be to use an even wider kernel to completely eliminate the ghost cell updates, however that would require increasing the number of ghost cells used by FsGrid. We choose to limit the number of ghost cells to four per dimension (2 ghost cells per side) to avoid the extra memory penalty and thus we limit ourselves to using 5-point stencils. A larger ghost domain would also make existing ghost communication more expensive. Furthermore, the memory footprint is the same for both methods and insignificant compared to the memory needed to store the velocity distribution function for each spatial cell, as shown in Table 1. The filtering operator presented in this work has been used to aid in 6D simulations performed with Vlasiator in efficiently alleviating the artifacts introduced by the staircase effect.



*Code and data availability.*   The Vlasiator simulation code is distributed under the GPL-2 open source license at https://github.com/fmihpc/vlasiator. In Vlasiator 5.0 (Alfthan et al., 2020) spatial AMR was introduced to enable the 6D simulations. The spatial filtering method as discussed in this work was introduced in Vlasiator 5.1 (Pfau-Kempf et al., 2021). The more efficient triangle filtering operator was introduced in Vlasiator 5.2.1 (Pfau-Kempf et al., 2022). The Analysator software (https://github.com/fmihpc/analysator, Battarbee and the Vlasiator team, 2020) was used to produce the presented figures. Data presented in this paper can be accessed by following the data policy
on the Vlasiator web site.

*Author contributions.*

KP carried out most of the study and the writing of the manuscript. MB, YPK, UG participated in the development of the numerical method presented in this work. MP is the PI of the Vlasiator model. All co-authors participated in the discussion of the results and contributed in improving the manuscript.

*Competing interests.*

The authors declare that they have no conflicts of interest.

*Acknowledgements.*   The simulations presented in this work were run on the LUMI supercomputer (Palmroth and the Vlasiator team, 2022) during its first pilot phase and the results analysis was done on the "Vorna" Cluster of the University of Helsinki. We acknowledge the European Research Council for Starting grant 200141–QuESpace, with which Vlasiator was developed, and Consolidator grant 682068–
PRESTISSIMO, awarded to further develop Vlasiator and use it for scientific investigations. The CSC – IT Center for Science in Finland and the PRACE Tier-0 supercomputer infrastructure in HLRS Stuttgart (grant no. 2019204998) are acknowledged as they made these results possible. The authors wish to thank the Finnish Grid and Cloud Infrastructure (FGCI) for supporting this project with computational and data storage resources. The Academy of Finland (grant nos. 312351, 328893, 322544, 339756, and 338629) is acknowledged.



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







**Figure 3.** Visualization of a 6D Vlasiator simulation with heterologous grids. a). Equatorial colour map of mass density (on the uniform FSGrid). b). Mass density along the profile shown as a dashed line in panel (a). c). Electric field magnitude colour map (on the uniform FSGrid). d). Electric field magnitude along the profile shown as a dashed line in panel (c). e) Magnetic field magnitude colour map (on the uniform FsGrid). f). Magnetic field magnitude along the profile shown as a dashed line in panel (e). The colour shaded regions in panels (b), (d) and (f) correspond to the resolution ratio between the field solver grid and the Vlasov grid. Unphysical oscillations in the field quantities, triggered by discontinuities in the moments, can be observed in the green and red regions.




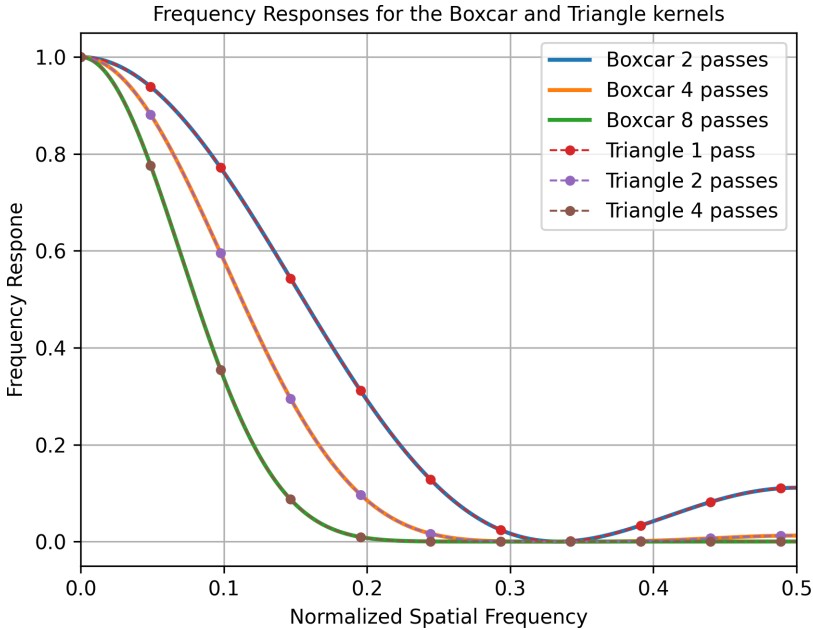

**Figure 4.** Solid lines: The frequency response of the boxcar filter operator for different numbers of passes as used in Vlasiator. Dashed lines: The frequency response of the triangular filter operator for different numbers of passes as used in Vlasiator.

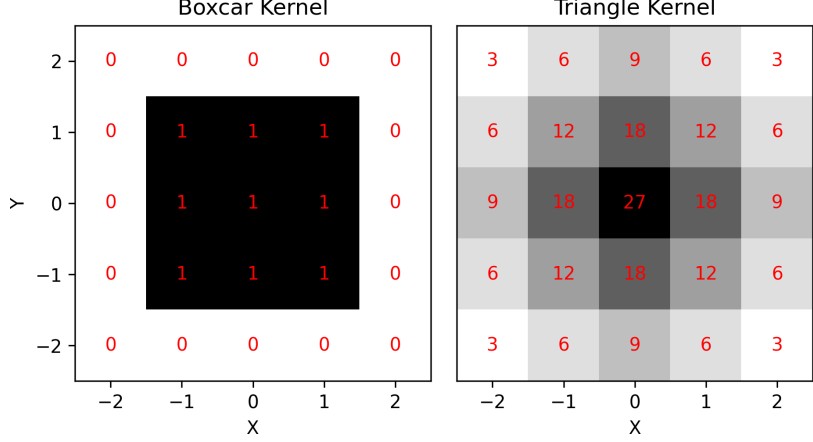

**Figure 5.** Graphical representation of the boxcar and the triangle kernels. The 2D slices are taken from the middle of the kernel's third dimension. For better clarity, the boxcar kernel is padded with zeros and the kernel weights, shown in red font, are not normalized.





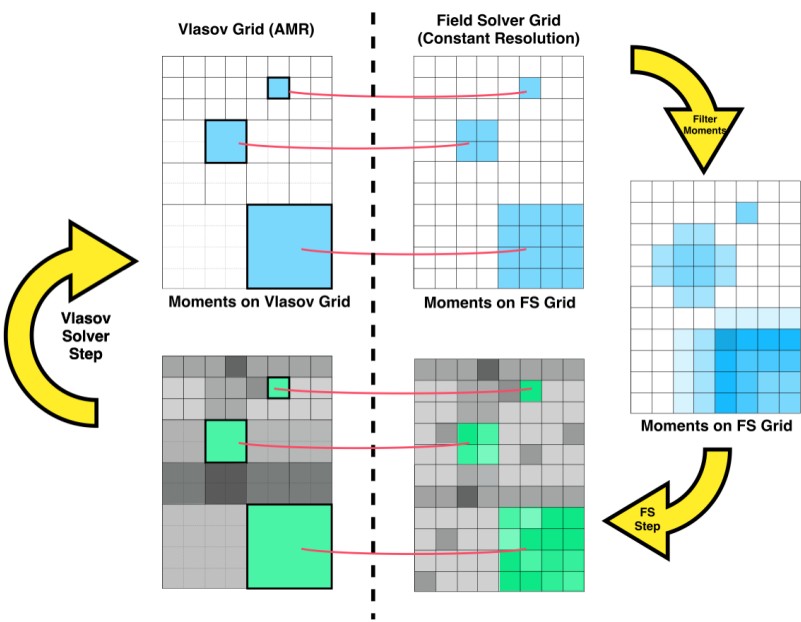

**Figure 6.** Schematic of Vlasiator's grid coupling scheme supplemented with the extra filtering step to alleviate the staircase effect in 6D AMR runs.

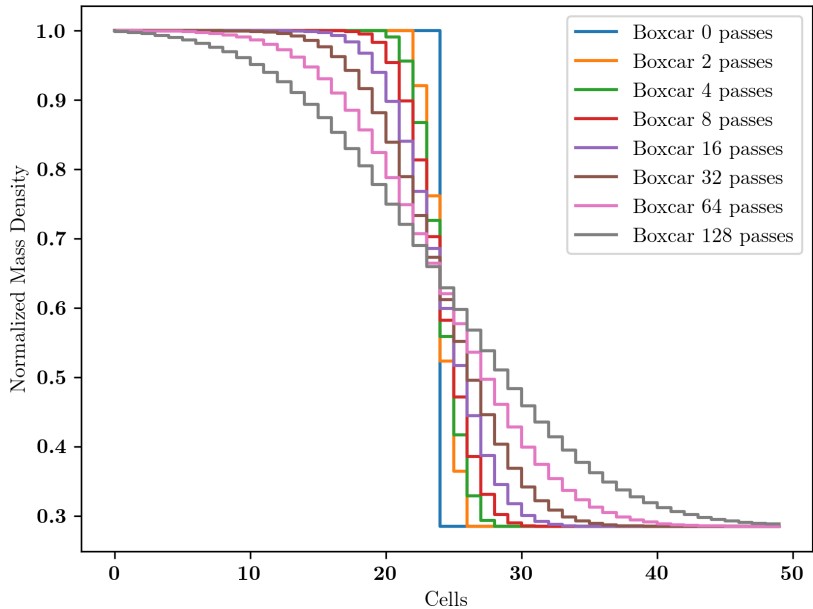

**Figure 7.** Smoothing of a step in mass density along a line in the simulation domain intersecting a discontinuity. The unfiltered normalised mass density line profile is shown in blue, and the other colours correspond to an increasing number of boxcar passes.





**Figure 8.** Vlasiator magnetospheric simulation results. a) Mass density colour map (on the uniform FSGrid) of 6D AMR simulation without the filtering operator. b) Mass density colour map (on the uniform FSGrid) of 6D AMR simulation with the filtering operator in use. There are four refinement levels and they are treated with 0,2,4,8 filtering passes from finest to coarsest respectively. The insets show a zoomed-in region from the bow shock. The FsGrid mesh is denoted by the black lines whereas the DCCRG levels are visible from the step discontinuities visible in the inset in panel (a). c) Electric field magnitude colour map (on the uniform FSGrid) with the filtering operator in use. d) Electric field profile sampled along the dashed line in panel (c). e) Magnetic field magnitude colour map (on the uniform FSGrid) with the filtering operator in use. f) Magnetic field profile sampled along the dashed line in panel (e).



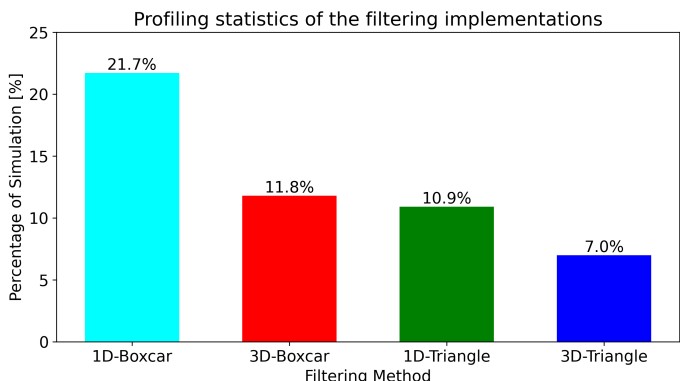

**Figure 9.** Profiling statistics for the different implementations of the filtering operators in Vlasiator. The results are derived from a custom simulation setup designed to test filtering performance.