# Peer review of "Spatial filtering in a 6D hybrid-Vlasov scheme for alleviating AMR artifacts: a case study with Vlasiator, versions 5.0, 5.1, 5.2.1"

_EGUsphere, 2022_

## Author Response (AR1)

**Response to Executive Editor**

**September 7, 2022**

*In the "Code and Data Availability section" of your manuscript, you state that the code for your work is archived on GitHub. Admittedly, later in the references, a ZENODO repository is cited, as you addressed the previous comments by the Topical Editor. However, the mentioned section must contain this information about the permanent repository. Therefore, please, in a potential reviewed version of your manuscript modify the 'Code and Data Availability' section, including the DOI of the Zenodo repository.*

We thank the executive editor for reading our manuscript and pointing out that our permanent repository is not included in the 'Code and Data Availability' section. We have revised that section in our udpated version of the manuscript to include our Zenodo reference DOI as well.

**Response to Reviewer 1**

**September 7, 2022**

The authors thank Reviewer #1 for reading our manuscript and providing useful and constructive comments. Below we list our point-by-point response to the Reviewer's comments. Italics are used to list the reviewer's comments and our response is in regular font.

*Below the authors list the reviewer's major comments on the manuscript.*

1. *Fig. 3 and Fig. 8 show numerical artefacts in XY plots but they do not show the AMR mesh. It is not clear how these are related exactly. The MS mentions the AMR mesh can be recovered from these figures (line 143). No, that is not enough. Please show the AMR mesh explicitly.*

   We agree that the wording used in Line 143 ( *" . . . we demonstrate results of the boxcar filtering operator in a large magnetospheric production scale run with four 4 AMR levels as illustrated in Figure 8."*) was unclear. Figures 8(b,c,e) are quantities on the field solver grid which is uniform. We have modified the manuscript and now the text reads (*" . . . we demonstrate results of the boxcar filtering operator in a large magnetospheric production scale run using four AMR levels. Simulation quantities on the refined AMR mesh are illustrated in Figure 8a and on the uniform field solver grid in Figures 8(b,c,d)."*). To address the raised point, we have supplemented Figure 1 with an explicit 3D plot of the AMR Vlasov grid, as the reviewer suggested.

2. *Also, why do the authors show smoothed ("fixed") solutions for Fig.8 but not Fig.3? This is confusing. If you show numerical issues, it makes sense to show how you fix them everywhere.*

   The aliasing effects on the field-solver quantities illustrated in Figure 3 require a lot of simulation time to propagate into the field solver. The simulations snapshots in Figure 3 are at t=1096.5 simulation seconds. Vlasiator is a very computationally expensive code and reproducing the simulation shown in Figure 3 with the filtering operator active would require large amounts of computational time, for which we do not currently have a Tier-0 computational grant. The aliasing results can be qualitatively compared between different 3D simulations, as shown in Figures 3(d,f) and Figure 8(d,f). As mentioned in the manuscript it is visible that the profiles in Figures 8(d,f) are not showing unphysical oscillation

unlike their counterparts in Figures 3(d,f) where the filtering operator is not used.

3. *What region exactly do those insets in Fig. 8 represent? Why does one show coarse cells and the other shows fine cells while they seem to represent the same AMR mesh with and without filtering?*

The spatial coordinates of the insets are exactly the same in Figure 8(a) and Figure 8(b). Both insets show the mass density at a zoomed in region close to the Earth's bow-shock. However, in the revised version of our manuscript Figure 8(a) illustrates mass density on the AMR Vlasov grid while Figure 8(b) illustrates mass density on the uniform field solver grid. The staircase effect is visible at the interfaces of the AMR refinement levels in Figure 8(a), since the filtering operator is not applied. In Figure 8(b) the filtering operator is applied and the staircase effect is alleviated. Based on the reviewer's comment this was not clear enough in the text, so we have modified the caption of Figure 8 to also read "...*The insets in panels (a,b) show a common zoomed-in region from the bow shock.*"

4. *What sense does it make to show Fig. 8(d,f) if they cannot be compared to similar unfiltered profiles?*

Our goal in showing Fig8(d,f) is to point that with the filtering operator enabled, the quantities on the uniform field-solver grid do not suffer from the high frequency oscillations that their counterparts in Fig3(d,f) suffer due to the staircase effect. We think that even though the two simulations are indeed different, a qualitative comparison of the electric and magnetic field profiles in Figure 3(d,f) and Figure 8(d,f) is enough to gauge whether the staircase effect is alleviated or still induces artifacts in the field solver.

5. *Given all the above, please show unfiltered and filtered plots side by side for comparison, together with the AMR mesh. Fig 3 and Fig.8(c-f) do not carry much information unless you show their counterparts next to them.*

We want to clarify that as mentioned in the manuscript the filtering operator is only applied to the plasma moments and not to the magnetic/electric fields. Thus we cannot show what the reviewer asks for since unfiltered electric/magnetic fields do not exist is a simulation where the filtering operator is active. However, Figure 3 is provided as a point of comparison between a simulation that does not use the filtering operator and one that does as illustrated in Figure 8. We have also elected not to show the AMR mesh as it is too fine to allow for the evaluation of the plot and the mesh at the same time. However in the revised version of the manuscript the zoom-ins in Figure 8(a,b) have the AMR enabled Vlasov grid overlaid on them as a point of comparison.

6. *Overall, my major concern is that it is not clear at this point if the authors have "cracked" the problem or not. This spatial filtering may be good enough to regularize the bow shock boundary, but this procedure may*

*result in modifying the global solution considerably. The only way to verify that is to also show a reference solution on a fine uniform mesh. I don't see those. It looks to me that the MS shows either unfiltered AMR solutions or filtered AMR solutions, without comparing them side by side or showing uniform mesh solutions next to them.*

We thank the reviewer for this comment. We agree that the only way to verify the filtering mechanism would be to compare the filtered results with a simulation without the filtering which is also artifact-free. That would mean running a 3D simulation with no AMR, using the maximum spatial resolution (highest refinement level in the AMR runs demonstrated in the manuscript) in the whole simulation domain. We can extrapolate the computational requirements of such a reference solution. Based on the computational resources used for the simulation in Figure 8, the best case scenario without taking into account scaling performance, the same simulation with no AMR would require upwards of 350 MCPUh for a total of 500 simulation seconds. That would make the simulation about 22 times more expensive than the ones presented in the manuscript. To put this into perspective, Prace's EuroHPC JU Call for Proposals grants at most 306 MCPUh (`https://prace-ri.eu/hpc-access/eurohpc-access/eurohpc-ju-call-for-proposals-for-regular-access-mode/`) on LUMI which ranks 3rd at TOP500 as of June 2022(`https://www.top500.org/lists/top500/2022/06/`). Further, we wish to point out that the filtering mechanism is not applied to the highest resolution regions in the simulation which are the regions with high scientific importance.

*Below the authors list the reviewer's minor comments on the manuscript.*

1. *Lines 100-105: How is Eq.6 related to numerical filters actually used in the MS? Either strike this equation out or explain how it is supposed to be used. Is 'j' the imaginary unit? Is this formula supposed to be used in Fourier transforms? Show that exactly in Eq. 7.*

   We agree with the reviewer's comment that this equation is not aiding in the scientific quality of the manuscript and we have removed it.

2. *Line 5: strike out 'resolution induced'*

   The authors have removed "resolution induced" from the text as per the reviewer's suggestion.

3. *Line 14: rephrase with '3 dimensions . . . . space'.*

   Based on the reviewer's comment this sentence has been modified to:
   ". . . hybrid-Vlasov plasma simulation code that models collisionless plasma by solving the Vlasov-Maxwell system of equations for ion particle distribution functions on a six dimensional Cartesian mesh, representing three spatial and three velocity dimensions."

4. *Line 19: strike out 'which in practice'*

   Done, thank you.

5. *Line 21: there's no Gauss law here (divE= 4 pi rho)*

   The authors have modified the text accordingly and thank the reviewer for pointing this out.

6. *Lines 30-31: Lapenta (2012) is not appropriate for referencing the PIC method in general.*

   The authors have modified the citation and are now citing Nishikawa et al. 2021 as a review of the Particle-in-Cell method.

7. *Line 134: replace 'artificial step' by 'density step'.*

   The authors have modified the manuscript accordingly.

**Comment on the Semi-Lagrangian solver**

*Lines 39-40: I am actually surprised that Vlasiator uses a semi-Lagrangian scheme in configuration and velocity space. These schemes are known to enhance numerical diffusion due to their inherent need to map Lagrangian particles back to the mesh. Please comment on diffusion effects they cause, compared to high-order Eulerian approaches. I understand that the Lagrangian step preserves positivity and is conservative. However, it is highly diffusive too, which is not discussed here.*

As mentioned in the manuscript Vlasiator uses the SLICE-3D algorithm to solve the Vlasov equation. Since this is a semi-Lagrangian solver it does not need to abide by the CFL limit which in essence means that less total steps are required to propagate the plasma compared to a Eulerian approach. Since diffusion is accumulated over many simulations steps this aids in keeping the total diffusion low. Moreover, Vlasiator is using slope limited polynomials for the mapping instead of tracking Lagrangian quasi-particles and that does not exactly compare to common SL approaches. Finally, Vlasiator uses a $5^{\text{th}}$ order polynomial reconstruction to perform the Lagrangian mapping which is accurate enough to ensure low diffusivity.

**Response to Reviewer 2**

September 7, 2022

*This paper designs some kernels to filter the staircase effects arising from AMR, which is innovative. The authors carefully examine the mass conservation and computational overhead. Great work!*

We thank the reviewer for his comments on our manuscript.

*We encourage the authors to check WarpX's ( `https: // warpx. readthedocs. io/ en/ 21. 02/ theory/ amr. html` work to see if any techniques related to the absorbing layers can be utilized. Also, moving the codes to the GPU architecture is another trend.*

We have modified the introduction to cite WarpX as related work. However, as also mentioned in the revised version of our manuscript, WarpX's methods are not compatible with Vlasiator since Vlasiator does not use a particle approach to modelling plasmas and also because Vlasiator's field solver operates on a uniform mesh.

---

## Author Response (AR2)

**Final Response**

October 1, 2022

We wish to thank both Referee 1 and Referee 2 for their constructive comments helping to improve the quality of our manuscript. We also wish to thank the Topical Editor for also contributing to the scientific quality of our work. In response to the Topical Editor's suggestions we have modified Figures 3 and 8 in the manuscript by enlarging the font used for the axis labels, ticks and titles in order to improve their readability. We have also removed the scientific notation from both Figures' colorbar labels and also scaled mass density from $kg\,m^-3$ to $m_p\,cm^-3$ where $m_p$ stands for proton mass, to further improve the readability of the Figures.